# Broad-Spectrum Efficacy and Modes of Action of Two *Bacillus* Strains against Grapevine Black Rot and Downy Mildew

**DOI:** 10.3390/jof10070471

**Published:** 2024-07-09

**Authors:** Robin Raveau, Chloé Ilbert, Marie-Claire Héloir, Karine Palavioux, Anthony Pébarthé-Courrouilh, Tania Marzari, Solène Durand, Josep Valls-Fonayet, Stéphanie Cluzet, Marielle Adrian, Marc Fermaud

**Affiliations:** 1National Research Institute for Agriculture, Food and the Environment (INRAE), Institute of Vine and Wine Sciences (ISVV), UMR Santé Agroécologie du VignoblE (SAVE), 71 Avenue E. Bourlaux, CS 20032, 33882 Villenave d’Ornon, France; 2Agroécologie, National Research Institute for Agriculture, Food and the Environment (INRAE), Institut Agro Dijon, Univ. Bourgogne, 21000 Dijon, France; 3Univ. Bordeaux, Bordeaux INP, National Research Institute for Agriculture, Food and the Environment (INRAE), OENO, UMR 1366, Institute of Vine and Wine Sciences (ISVV), 33140 Villenave d’Ornon, France; 4Bordeaux Metabolome, MetaboHUB, 33140 Villenave d’Ornon, France

**Keywords:** *Bacillus velezensis*, *Bacillus ginsengihumi*, biocontrol, biological control, *Vitis vinifera*, *Guignardia bidwellii*, *Plasmopara viticola*, antibiosis, plant defence

## Abstract

Black rot (*Guignardia bidwellii*) and downy mildew (*Plasmopara viticola*) are two major grapevine diseases against which the development of efficient biocontrol solutions is required in a context of sustainable viticulture. This study aimed at evaluating and comparing the efficacy and modes of action of bacterial culture supernatants from *Bacillus velezensis* Buz14 and *B. ginsengihumi* S38. Both biocontrol agents (BCA) were previously demonstrated as highly effective against *Botrytis cinerea* in grapevines. In semi-controlled conditions, both supernatants provided significant protection against black rot and downy mildew. They exhibited antibiosis against the pathogens by significantly decreasing *G. bidwellii* mycelial growth, but also the release and motility of *P. viticola* zoospores. They also significantly induced grapevine defences, as stilbene production. The LB medium, used for the bacterial cultures, also showed partial effects against both pathogens and induced plant defences. This is discussed in terms of choice of experimental controls when studying the biological activity of BCA supernatants. Thus, we identified two bacterial culture supernatants as new potential biocontrol products exhibiting multi-spectrum antagonist activity against different grapevine key pathogens and having a dual mode of action.

## 1. Introduction

Grapevines (*Vitis vinifera* L.), like other cultivated plants, can be infected by a wide variety of microorganisms including notably oomycetes and fungi. These mostly include downy mildew (DM), powdery mildew, grey mould, and black rot (BR). Grey mould is caused by the necrotrophic ascomycete *Botrytis cinerea* Pers.:Fr., an ubiquitous plant pathogen with numerous host plants [1,2]. Downy mildew is caused by *Plasmopara viticola* ([Berk. and M.A. Curtis] Berl and De Toni), an obligate biotrophic oomycete. In successive sexual and asexual cycles, zoospores penetrate via stomata and develop intercellular mycelium with haustoria. Upon high hygrometry, sporulation occurs with the emergence of sporangiophores, through stomata, producing sporangia that will release zoospores and initiate secondary infections (for review—[3,4]). DM is present in almost all vineyards worldwide. Although historically BR was not so frequent in the past decades [5], it may cause disastrous losses in North America and Europe [6,7], and is in resurgence in several countries [8]. The causal pathogen is *Guignardia bidwellii* (Ellis) Viala and Ravaz (anamorph: *Phyllosticta ampelicida*), a hemibiotrophic ascomycete [9]. Primary infections, generally occurring on leaves during spring, mostly originate from ascospores—but also possibly from pycniospores—released from overwintered mummified berries [10]. They germinate and develop an appressorium, from which one or two infection hyphae penetrate through the cuticle and then colonize the plant organ by subcuticular hyphae [11,12,13]. After an incubation and latency period, pycnidia are produced within typical circular necrotic spots on the upper leaf side [14]. After asexual cycle(s) on leaves, the released pycniospores will initiate secondary infections in inflorescences and bunches, that may also still be infected by ascospores. When symptomatic, berries wilt and develop as mummified with a typical purplish-blue colour. Dark pycnidia and pseudothecia form on them and overwinter, thus constituting a source of inoculum for primary infection at the beginning of the following year [15].

Protecting vines against all these diseases is essential to maintain yield and prevent quality losses [16]. If weather conditions are favourable, numerous treatments are necessary: e.g., up to 15 or more against DM with contact fungicides. To date, protection strategies mainly rely on the use of chemical fungicides. However, many of them cause health [17,18] and environmental problems [19,20], can lead to the selection of fungal-resistant strains [21] and leave residues on grapes and in wines [22]. Organic and/or biodynamic viticulture may offer solution(s), but the copper-based fungicides used continuously increase copper levels in the soil, exacerbating its impact on microorganisms and plants [23]. An alternative strategy is breeding. New varieties resistant to downy and powdery mildews have been generated by introgression of resistance genes. For example, four varieties, including Artaban, have been created and registered in France, in 2018 [24]. Nevertheless, the resistant varieties remain susceptible to other diseases, notably BR and with a very high level of susceptibility to this disease. In addition, and to also preserve sustainable resistance, specific fungicide treatments are hence still necessary [9,25]. Finally, since the deployment of resistant varieties remains limited, the use of complementary solutions to reduce the use of chemical fungicides is essential for the protection of traditional cultivars. Among them, bioprotection, of which biocontrol is already in use, deserves to be implemented more extensively to establish sustainable production systems [26].

Bioprotectants active against fungal diseases include algae, plant and microorganism extracts, biocontrol agents (BCA) and the active molecules they secrete. BCA act through various modes of actions, such as antibiosis, niche or nutrient competition, and/or the stimulation of plant defences. Plants indeed have an immune system that enables them to perceive microbial molecular patterns (MAMPs) or damage-associated molecular patterns (DAMPs), owing to pathogen recognition receptors (PRR) located at the plant cell plasma membrane [27,28,29]. In grapevines, we have investigated some of these receptors [30] and identified those for the bacterial flagellin [31] and chito-oligosaccharides [32]. Recognition of MAMPs/DAMPs by PRR activates defensive responses. These involve a cascade of signalling events, leading to the activation of defence genes. This activation leads to defensive responses: the synthesis of PR proteins, phytoalexins, and cell-wall reinforcements [30]. These defences can also be induced by active compounds secreted by microorganisms, generally obtained in their culture filtrates, but also by microbial extracts [33,34]. Stilbenes represent grapevine phytoalexins, with resveratrol being the monomer unit. Their role in the grapevine defence against various key pathogens including *P. viticola* and *B. cinerea* [35] has been reported.

Bioprotectants were sought to protect grapevines against cryptogamic diseases [36]. BCA and their excreted molecules were reported to be active, mainly against necrotrophic pathogens, particularly *B. cinerea*, and some causing grapevine trunk diseases (GTD) pathogens, notably *Neofusicoccum parvum* [37,38,39,40,41,42]. However, this was investigated to a much lesser extent against biotrophic mildews and/or hemibiotrophic BR pathogens. A few bioprotection studies have been carried out against BR, notably by using plant saponin-containing extracts, but without any conclusive efficacy results in the vineyards [13,15,43]. Some very encouraging results were nonetheless obtained, notably including the activity of GC-3 (a combination of cottonseed, corn oil and garlic extract—[44]). Regarding BCA, the efficacy of Serenade (*Bacillus subtilis*) was also better documented but was highly dependent upon the tested formulation [44,45,46].

More specifically, among bacterial BCA, the genus *Bacillus* is largely used in numerous crops [33,47]. In grapevines, we have reported the activity of *B. subtilis* strain GLB191 against DM [33]. The *Bacillus* strains *B. velezensis* Buz14 (isolated from peach) and *B. ginsengihumi* S38 (isolated from grapevine wood) have been compared and proved highly effective to protect table grapes (cv. Thomson Seedless) against *B. cinerea*. Their modes of action were mostly elucidated, except for defence gene elicitation [40,41,48]. Notably based on different antibiosis abilities, these modes of action may account for a quite broad-spectrum control of diseases due to various necrotrophic pathogens. The Buz14 strain was hence of interest in different postharvest pathosystems (orange, apple, grape and stone fruit), with *Penicillium digitatum*, *P. expansum*, *P. italicum*, *B. cinerea*, *Monilinia fructicola* and *M. laxa* [49]. As for the S38 strain, such a broad spectrum activity against grapevine necrotrophic GTD pathogens was less noticeable, i.e., medium-low efficacy (between 20 and 30%) against *Phaeomoniella chlamydospora* and *Neofusicoccum parvum* [37,50].

In this context, the aim of this study was (1) to test and compare, in semi-controlled conditions, the efficacy of the culture supernatants of the two experimental *Bacillus* strains, Buz14 and S38 strains, against key grapevine pathogens, i.e., *P. viticola* (biotrophic) and *G. bidwellii* (hemibiotrophic); and (2) to study and compare their modes of action, according to the highly different lifestyles and infection pathways of the two pathogens. We observed that the culture supernatants of both bacterial strains are active against DM and BR. The activity is due to excreted bacterial metabolites together with components of the culture medium. Their modes of action combine antibiosis and activation of grapevine defence responses (stilbenes).

## 2. Materials and Methods

### 2.1. Biological Material

#### 2.1.1. Grapevine Cuttings Production

For *in planta* bioassays, one *Vitis vinifera* cultivar (cv.), Marselan (Cabernet Sauvignon × Grenache), and the hybrid variety Artaban, which is part of the INRA-ResDur breeding program [51], were used. Grapevine plants were issued from wood cuttings grown in the greenhouse in 9 × 9 cm individual pots, containing a mixture of peat and perlite (Substrate5—Recipe 446, Klasmann-Deilmann, Ruy-Montceau, France). They were exposed to a 16 h photoperiod [52], with an additional artificial lighting system (Na-lamps—350 μmol.m^−2^.s^−1^) when the natural photoperiod was too short (from January to the end of March).

Until they developed eight to ten fully expanded leaves, plants were watered every two days and fertilized weekly (Allrounder 20/20/20, ICL France, Limas, France). In order to prevent powdery mildew, sulphur evaporators were installed (HotBox Sulfume, Growland, Hamburg, Germany). All the cuttings were finally washed three hours before the first treatment, so as to remove sulphur residues, which may have a side effect on BR [9,15].

For protection assays against DM and defence analysis experiments, Marselan plants were cultivated in another site (INRAE Dijon). Plants obtained from herbaceous cuttings were grown in a greenhouse in individual pots (8 cm × 8 cm × 8 cm) containing a mixture of peat and perlite (7:3 *v/v*) at 23/15 °C (day/night), under a 16 h light photoperiod. Plants were watered with a nutritive solution (N/P/K 10-10-10, Plantin, France) and were used for experiments when they developed 5 to 7 fully expended leaves.

#### 2.1.2. Pathogen Strains and Culture Conditions—Inoculum Preparation



*G. bidwellii*



Two *G. bidwellii* strains, namely GF2 (isolated from foliar symptoms—Villenave d’Ornon, France) and 111645 (f. sp. *parthenocissi*, obtained from the Fungal Biodiversity Centre—[53]) were cultured for the assays. Both strains were maintained on oat-meal agar—2% oatmeal and 1.5% agar—culture medium in Petri dishes [54], at 23 °C.

For *in planta* bioassays, spore suspensions were prepared as follows:-Asexual inoculum: after a 3-week growth period under permanent light at constant 23 °C in a growth chamber (LMS 610XAP, LMS Ltd., Sevenoaks, Kent, United Kingdom), pycniospores were obtained by flooding the Petri dishes with 10 mL of sterile deionised water, and the spore concentration was further adjusted by hematocytometry (see also hereafter) at the desired density, 2 × 10^4^ spores.mL^−1^ [12,13].-Sexual inoculum: overwintered fruit mummies (cv. Prior) were used, originating from an INRAE vineyard near Bordeaux (Château Couhins, Gironde, France) in fall in 2021 and 2022, and intended for use the following year. Mummies were stored outdoors for perithecia maturation during winter. At maturity, spores were obtained by soaking mummies in sterile water for 30 to 45 min [10]. In the spore suspensions, ascospore concentration was adjusted at the desired density, namely 2 × 10^4^ spores.mL^−1^ of suspension, by hematocytometry.



*P. viticola*



The *P. viticola* strain used was collected in a Burgundy vineyard (SRPV, Beaune, France) and maintained on Marselan plants as previously described [55]. Briefly, abaxial sides of leaves were sprayed with a suspension of 10^4^ sporangia.mL^−1^ in ultrapure water. Inoculated plants were placed in a humidity chamber (relative humidity (R.H.) > 95%, 21 °C in darkness) for at least 6 h, to facilitate the encystment before being transferred back to the greenhouse under the conditions indicated above. Six days after inoculation, plants were again placed overnight in the humid chamber to induce sporulation.

#### 2.1.3. Bacterial Supernatant Production

Two *Bacillus* strains, namely Buz14 (*B. velezensis*) and S38 (*B. ginsengihumi*—[48]), were selected and evaluated for their potential antifungal activity against the different plant pathogens.

These two strains were first grown in the dark for 48 h, in 3 L Erlenmeyer flasks and under magnetic stirring (180 rpm—27 °C), in Luria–Bertani (LB) liquid culture medium (Fisher Scientific, Hampton, NH, USA). LB culture medium (tryptone, 10.0 g; yeast extract, 5.0 g; sodium chloride, 10.0 g) was prepared by dissolving 25 g of the product powder in 1 L of deionized water and autoclaved (121 °C—20 min) before use. At the time of harvesting, bacterial growth was determined as previously described [37].

Culture supernatants were then obtained by two successive centrifugation steps, for 10 min at 5100 rpm. The supernatants were later collected and stored at −20 °C until further use.

LB medium alone was incubated under the same conditions and used as a control in all following assays.

### 2.2. In Planta Protection Assays

#### 2.2.1. Black Rot Experimental Design and Disease Evaluation

Overall, two different experiments were carried out during the first year, i.e., 2022, and three successive ones during the second one, i.e., 2023. In every experiment, within the same grapevine variety, plants were arranged in a completely randomized block design, with 3 blocks each, and 2 to 3 replicates per block.

Leaves were sprayed (WS2342 spray, WESCO, France) until run-off, at a volume of 10 mL of solution per cutting, under the following conditions: supernatants were applied once (48 h or 24 h prior to inoculation) at different concentrations (25 and 50% of supernatants in aqueous solution), and so was the LB medium alone (25 and 50% in aqueous solution). The control plants were sprayed using tap water only.

Cuttings were then inoculated, by spraying either spore suspension (as described in Section 2.1.2) until run-off on the adaxial face of potentially susceptible leaves, including the apex and the first six fully expanded leaves [13]. A 10 mL small glass hand atomizer was used [12]. Additionally, the first expanded leaf (L1, displaying a surface between 4 and 9 cm^2^) was identified using a coloured plastic tag.

Directly following inoculation, cuttings were placed for 24 h under plastic domes, to maintain a high R.H. (between 90 and 100% R.H.), mandatory for spore germination and pathogen penetration [9,56].

After a two-week incubation period, when typical BR symptoms were expressed and fully developed—that is, foliar symptomatic spots showing visible pycnidia and the specific surrounding brownish dark edge [9]—disease severity was assessed on every cutting by visually measuring the BR typical lesion surface on the symptomatic leaves. The six (upper) youngest fully expanded leaves at the time of inoculation, together with the first two leaves that developed after inoculation, were taken into account for the evaluation. For each plant, disease severity was further calculated as a mean value of the eight leaves considered, as previously described [13], and expressed on a relative scale, as a percentage in relation to the leaf surface at each leaf stage, taken individually. Two or three independent biological repetitions were performed.

#### 2.2.2. Protection Assays against Downy Mildew

Supernatants (10 and 25%) were sprayed on leaves until run-off. Ultrapure water and LB medium (10 and 25%) were used as controls. After allowing treatments to dry for at least 2 h at room temperature, plants were placed in the greenhouse, under the conditions indicated above (Section 2.1.1). Two days after treatment, abaxial faces of treated leaves were inoculated with a sporangia suspension (10^4^ sporangia.mL^−1^). Inoculated plants were placed in a humid chamber (R.H. > 95%) in darkness for at least 6 h before moving back to the greenhouse. Five days after inoculation, leaf discs (1.3 cm diameter) were punched out and placed with the abaxial side uppermost, on Whatman paper soaked with water, in a closed plastic box. This system was left overnight in darkness at 21 °C and saturated relative humidity to trigger sporulation. Finally, photos of the discs were taken before being analysed with the image analysis software Visilog 6.9 (Noesis, Paris, France; [57]). Fifteen leaf discs from 3 plants per condition were used, and three independent biological repetitions were performed.

### 2.3. Evaluation of Direct Toxicity of the Bacterial Supernatants towards Pathogens

#### 2.3.1. *In Planta* Experiments

Towards *G. bidwellii*

For *in planta* antibiosis bioassays against *G. bidwellii*, two different experiments were carried out during the first year, i.e., 2022, and two successive ones during the second year, i.e., 2023. In every experiment, within the same grapevine variety, plants were arranged in a completely randomized block design, with 3 blocks each, and 2 to 3 replicates per block. Two or three independent biological repetitions were performed.

Experimental design and symptom evaluation were carried out as described in Section 2.2.1, with slight modifications, as follows:

Leaves were sprayed 2 h prior to inoculation with the supernatants, LB medium, or tap water (control), and at a single concentration (25% of supernatants or LB in aqueous solution).

Cuttings were then inoculated by spraying an ascospore suspension (as described in Section 2.1.2) only, until run-off.

Towards *P. viticola*

Leaves were treated as described in Section 2.2.2 and inoculated (10^5^ sporangia.mL^−1^) 2 h after treatment. Twenty-four hours after inoculation, 10 leaf discs (0.8 cm diameter) were punched out before being immersed in absolute methanol for at least 24 h. This step allows fixing of the plant tissues and solubilizing of the chlorophyll pigments. Then, the discs were transferred to chloral hydrate (1 g.mL^−1^) until complete transparency (overnight). After rinsing with a phosphate buffer (0.1 M, pH 9), discs were immersed in aniline blue solution overnight [58]. Infection sites were shown by epifluorescence microscopy [λ exc 340–380 nm, λ em 425 nm (long pass filter)]. Three representative fields per disc were observed. Three independent biological repetitions were performed.

#### 2.3.2. *In Vitro* Experiments

Supernatant anti-germinative activity against *G. bidwellii* pycniospores

Germination bioassays were performed on collodion membranes placed over a water agar medium, as previously described [13]. Briefly, collodion membranes were prepared by pipetting 45 µL of collodion (4%, 1026441000, Merck, Saint Quentin Fallavier, France) directly onto a microscopic slide. Once the solvents had evaporated, microscopic slides coated with collodion were placed in a water-filled beaker, so as to float the membranes. These latter were then collected and placed over a water agar medium (1.5%), in 9 mm Petri dishes.

Each membrane was further inoculated using a mix of 5 µL of pycniospore suspension (as previously described in Section 2.2.1) and 5 µL of the solution to test (control, LB, supernatants, or Polyram^®^, BASF, Ecully, France – used at 0.2% as a positive control), and each treatment consisted of 5 membranes (replicated 3 times). The final concentrations of the solutions to test were 50% (supernatants and LB). To ensure optimal germination conditions for the spores [59], Ca^2+^ was added in the spore suspension (10 mM), and pH was checked and adjusted in both spore suspension (6.5) and water agar medium (6).

After inoculation, Petri dishes were sealed and placed at 23 °C, in the dark, for 16 h. At the end of this incubation period, at least 25 spores were examined per membrane and classified according to their development stage, following recommendations [13]. Mean germination rate was finally estimated by comparing the number of germinated spores (“with germ tube” and “with germ tube and appressorium”) to the total value of examined spores, for each experimental condition. Three independent biological repetitions were performed.

Supernatant antifungal activity against *G. bidwellii* mycelial growth

Antifungal activity of the supernatants against *G. bidwellii* was evaluated by means of a direct contact *in vitro* bioassay. An oatmeal agar medium was first prepared (2% oat flakes and 1.5% agar), in which the supernatants were mixed, before agar medium solidification, so as to obtain a final concentration scale ranging from 0.02 to 25% of supernatant in the medium, and poured into 9 mm Petri dishes. Discs of *G. bidwellii* (0.6 cm) were then cut out from the periphery of a 21-day-old fungal colony, and 3 discs were placed per Petri dish, containing the agar medium complemented with the supernatant.

Petri dishes were then sealed and incubated for 14 days in the dark, at 23 °C. At the end of this incubation period, mycelial radial growth was measured (two orthogonal measurements per fungal colony), and the inhibition rate was calculated for each product concentration, using Equation (1):(1)$$\frac{X_{0}-X_{t}}{X_{0}}\times 100$$
where X_0_ and X_t_ stand for the average diameter of the fungal colony in control and in treatments, respectively.

The test conditions consisted of aqueous solutions of both supernatants (Buz14 and S38), as well as LB medium alone. Sterile water was used as a negative control, while a marketed BCA, Sonata^®^ (*B. pumilus* QST 2808, Bayer Crop Science, Lyon, France), and a commercial fungicide, Polyram^®^ (Metiram, BASF), were used as positive controls. Each treatment was replicated five times.

The half-maximal inhibitory concentration (IC_50_) of supernatants (expressed as a percentage) required to obtain a fungal pathogen growth inhibition of 50% was also calculated, based on the dose–response curve for each evaluated product (Appendix A). A graphical interpolation, complemented with a statistical analysis based on a nonlinear regression [60], was used to calculate the IC_50_ values. Three independent biological repetitions were performed.

Effects on the release and the motility of *P. viticola* zoospores

Supernatants at 25% were applied on a suspension of sporangia (10^5^ sporangia.mL^−1^). The whole was placed under agitation (200 rpm) at room temperature for 2 h (time necessary for the zoospore release from the sporangia). Then, the number of empty and full sporangia was counted using a Malassez cell under a light microscope (magnification ×100). Sporangia that were still full were a sign of inhibition of zoospore release from the sporangia.

To observe the effect of the treatment on zoospore mobility, the treatment was applied after the release of zoospores from sporangia. Only 2 min after treatment, microscopic observations of the Malassez cell and counting were performed. The counting was conducted in one rectangle (0.25 mm × 0.20 mm) of the Malassez cell and consisted of counting all the zoospores entering this rectangle during 1 min.

IC_50_ values were determined by probit regression analysis using percentage of full sporangia data.

Three independent biological repetitions (with 3 technical replicates for each) were performed.

### 2.4. Assessment of Defence-Related Responses

Treated leaves (LB, Buz14 and S38 supernatants at 25% or water, as negative control) were collected at 10 h post-treatment (hpt) or 24 hpt for gene expression analyses or stilbene analyses, respectively. They were rapidly immersed in liquid nitrogen and grounded using TissueLyser II (Qiagen, Les Ulis, France). For stilbene analyses, samples were lyophilized.

#### 2.4.1. Gene Expression Analyses by qRT-PCR

Total RNAs were extracted from around 80 mg of leaf powder with a Spectrum Plant Total RNA Kit (Sigma, Saint Quentin Fallavier, France) according to the manufacturer’s instructions and including a DNAse step. Concentration and purity of RNA were assessed by spectrophotometry. One µg of total RNA was reverse-transcribed using the Superscript III Reverse Transcriptase kit (Thermofisher, Carlsbad, CA, USA) according to the manufacturer’s instructions. qRT-PCR experiments were performed as previously described [33]. Relative gene expression was calculated according to the Common Base Method [61], as described by [32]. Data were normalized with water control and housekeeping genes (*VvVATP16* and *EF1α*; Appendix A). Three independent biological repetitions (with 3 technical replicates per biological experiment) were performed.

#### 2.4.2. Stilbene Extraction and Analyses

A first extraction was made by adding 1 mL of methanol 100% to 50 mg of leaf powder. The resulting suspension was then vortexed and sonicated for 15 min. The sample was then centrifuged for 10 min at 10,000 rpm. Supernatant was recovered and a second extraction was performed from the resulting pellet with 1 mL of methanol 70%. Finally, the two supernatants were pooled and centrifuged for 5 min at 14,000 rpm at 4 °C before UHPLC-QqQ analysis. An aliquot of 500 µL of the extract was evaporated to dryness (SpeedVac) and resuspended in 100 µL of MeOH 50% for detection and quantification of stilbenes.

The samples were analysed by targeted analysis using high-performance liquid chromatography (Agilent Technology 1260 Infinity HPLC, Santa Clara, CA, USA) coupled with a triple quadrupole mass spectrometer (Agilent Technologies 6430 Triple Quadrupole detector). Chromatographic separation was conducted on an Agilent Zorbax SB-C18 column (100 mm × 2.1 mm, 1.8 µm, Santa Clara, CA, USA) at 40 °C with a binary solvent system composed of solvent A (0.1% formic acid in milli-Q water) and B (0.1% formic acid in acetonitrile LC-MS grade), with a flow of 0.3 mL/min. The gradient elution was set as follows: 0–4 min, 1–10% B; 4–12 min, 10–20% B; 12–13 min, 20–30% B; 13–16 min, 30% B; 16–18 min, 30–35% B; 18–20 min, 35–50% B; 20–23 min, 50–70% B; 23–24 min, 70–95% B; 24–26 min, 95% B; 27 min, 1% B). The injection volume of samples and standards was 4 µL. We used standard concentrations, between 0.0003 and 2.84 mg/L (for the 4 µL injection), to generate calibration curves for each stilbene compound studied and to quantify them in the samples, except as indicated in Appendix A.

For mass spectrometry analysis, the parameters used for the source were as follows: gas temperature, 350 °C; nitrogen flow rate, 11 L.min^−1^; nebulizer pressure, 15 psi; and capillary voltage, 3000 V. Measurements were made with a multiple reaction monitoring (MRM) method in positive or negative mode according to the compounds studied. Fragment conditions and collision energies had been previously published [62]. Finally, all the results are expressed as micrograms of compounds per gram of leaf (dry weight (DW)).

### 2.5. Statistical Analyses

Statistical analyses were performed using R statistical software version 4.3.2 (R Core Team, 2023, Vienna, Austria), and XLSTAT software (Version 2022.3.1, Adinsoft^©^, Paris, France).

For BR assays, disease severity data were first subjected to both the Shapiro–Wilk normality test and the Levene homoscedasticity test. To better fit these hypotheses, a variable transformation was used, namely log10(X + 1). Data were then subjected to either one-way or two-way ANOVA (at a 5% significance level), complemented with the Tukey Honestly Significant Difference (HSD) post-hoc test. Specific use of statistical tests is further detailed in the figure captions.

For DM assays and gene expression analyses, data normality (Shapiro test) and variance homogeneity (Levene test) were assessed at a significance level of 5%. Based on the results, treatments were compared statistically either parametrically using the one-way ANOVA test, or non-parametrically using the Kruskal–Wallis test at the 5% significance level. Where necessary, Tukey’s post-hoc test or Wilcoxon’s pairwise test with Holm’s correction were used at the same level of significance, respectively.

In graphs, significant differences between treatments are recognized if they have no letter (a, b, c, etc.) in common.

## 3. Results

### 3.1. Preventive Protection Assays

#### 3.1.1. Efficient Preventive Protection against *G. bidwellii*

Supernatant preventive application 24 h pre-inoculation

Based on a preventive application 24 h before inoculation (using pycniospores), both S38 and Buz14 supernatants were shown to provide a significant reduction in BR disease severity, when compared to both the water-treated control and the LB medium alone (Figure 1A). At a 50% concentration, both strains were able to significantly reduce disease severity by 90%, in comparison with the untreated control. For illustrative purposes, pictures of infected leaves, representative of the low and high severity levels displayed in the following figures, are presented in Appendix A.

At a lower concentration of supernatant (25%), both *Bacillus* strain supernatants provided high levels of protection (Figure 1A), with respective disease reductions of 52 and 57% (Figure 1A), for S38 and Buz14, when compared to the water-treated control.

Furthermore, the LB medium alone was able to significantly reduce disease severity when applied at a 50% concentration (77% reduction), while this effect disappeared at a lower dose.

Comparatively, when grapevine cuttings were inoculated using a sexual inoculum, i.e., ascospores (Figure 1B), high levels of protection were also demonstrated following a preventive spraying 24 h before inoculation. Both supernatants significantly reduced disease severity in comparison with the untreated control. The corresponding reductions reached 64 and 81%, for S38 and Buz14, respectively.

In addition, LB medium alone did not significantly reduce disease severity when compared to the untreated control, despite the 50% concentration that was applied.

Supernatant preventive application 48 h pre-inoculation

So as to evaluate the potential of the experimental BCA supernatants, an earlier spray timing was also evaluated (48 h before inoculation, without washout) by focusing on one of the *Bacillus* strain supernatants, S38, at a 25% concentration (Figure 2).

Regardless of the inoculum type used, the S38 supernatant application provided a highly significant disease reduction. When inoculations were performed using asexual (pycniospore) and sexual (ascospore) inocula, the protection efficacy reached 84 and 97%, respectively, when compared with the untreated control.

At this 25% concentration and whatever the inoculum used, it was also noticeable that the control LB medium, alone, significantly reduced the disease severity by about 16 and 62%, against pycniospore and ascospore infections, respectively.

#### 3.1.2. Supernatant Efficient Preventive Application against Downy Mildew

Based on a preventive application 48 h before inoculation, both supernatants used at 10% induced a significant decrease in *P. viticola* sporulation, when compared with the water control and LB medium alone (Figure 3). At a 25% concentration, the sporulation was almost completely inhibited (0.3 and 0.6% leaf sporulating area for Buz14 and S38, respectively). LB medium 25% also significantly inhibited sporulation, when compared to the water control (4.4 and 10.4%, respectively), but to a lesser extent than the supernatants. As the 25% treatment appeared to be the most effective one against DM, this concentration was kept for further analysis (i.e., direct effects against *P. viticola* and defence-related experiments).

### 3.2. Evidence of Direct Toxicity of the Bacterial Supernatants towards Pathogens

#### 3.2.1. *In Planta* Experiments

*In planta* direct toxicity towards *G. bidwellii*

Considering the biocontrol of the primary sexual inoculum of *G. bidwellii*, i.e., ascospores, the antifungal direct effect of the two BCA supernatants was shown *in planta* following a 2 h preventive treatment (Figure 4). Similar results were obtained on the two grapevine varieties, namely Marselan and Artaban, without any significant effect of the cultivar.

As shown in Figure 4, the Buz14 supernatant, at the 25% concentration, showed a highly significant reduction in BR severity (82%), when compared with the untreated control.

This clearly demonstrated a direct antibiosis effect, due to the presence of active antifungal secondary metabolites in the Buz14 supernatant.

Moreover, the LB medium at 25% also showed, by itself, a significant direct antifungal effect on the sexual infections by ascospores (64%), when compared to the untreated control.

*In planta* direct toxicity towards *P. viticola*

To assess the direct effect of BCA supernatants *in planta* against *P. viticola*, the encystment of zoospores (infection step) was studied on leaves inoculated only 2 h after treatment. Both bacterial supernatants, Buz14 and S38, and LB medium used at 25% significantly decreased the number of encysted sites compared to the water control (Figure 5). This direct effect was significantly stronger for BCA supernatants compared to LB (0.9 and 0.7 for S38 and Buz14, respectively, compared to 5.6 for LB medium), suggesting the presence of BCA-specific active metabolites. Finally, no statistical difference was found between the two bacterial supernatants.

#### 3.2.2. *In Vitro* Experiments

*In vitro* anti-germinative activity against *G. bidwellii* (pycniospores)

When *G. bidwellii* pycniospores were exposed to the *Bacillus* supernatants, at the 50% concentration, there was no significant decrease in the germination rates (Figure 6), whatever the tested BR strain (*Vitis*-specific strain, GF2; *Parthenocissus*-specific strain, 111645). Similarly, Sonata^®^ did not significantly reduce the germination rates either. This demonstrates that the supernatants could rather inhibit mycelial development or further penetration phases in plant tissues. In addition, on the GF2 strain, the control LB medium alone applied at 50% led to a significant reduction in the germination rate (68%), when compared with the untreated control. However, this was associated with highly variable data.

It should also be noted that pycniospore status was classified according to the development stage, i.e., “with germ tube” or “with germ tube and appressorium”. In all our experimental conditions, similar results were observed between the number of germinated spores (total), and the number of spores with germ tube and appressorium. This means that, when spores were able to germinate (after 16 h of experiment), an appressorium developed shortly afterwards, and that regardless of the strain and the treatment.

*In vitro* antifungal activity against *G. bidwellii* (mycelial growth) and *P. viticola* (zoospore-release inhibition)

Regarding *G. bidwellii* fungal growth inhibition, the two experimental BCA supernatants were able to significantly reduce the *in vitro* mycelial growth. The obtained IC_50_ ranged from 1.3 to 3.6% and 1.8 to 5.7%, with no significant difference, for S38 and Buz14 supernatants, respectively (Table 1). Additionally, similar results were obtained between the two fungal strains for both BCA supernatants. The dose–response curves, obtained with both *G. bidwellii* strains, are provided as Appendix A (*Vitis*-specific strain, Appendix A; *Parthenocissus*-specific strain, Appendix A).

Furthermore, interestingly, the control LB medium alone was also able to reduce the fungal growth, but to a significantly lesser extent compared with the bacterial supernatants. The related IC_50_ ranged from 39.8 to 64.1%, according to the strain effect that was observed with the LB medium only. In comparison, Polyram^®^ (a marketed synthetic fungicide) displayed a 100% inhibition at a 0.2% concentration, while Sonata^®^ (a marketed biocontrol product) displayed an IC_50_ of 1.6%, regardless of the *G. bidwellii* strain.

With regard to inhibition of zoospore release by *P. viticola* sporangia, the findings are very similar to those obtained for *G. bidwellii* (Table 1). Both BCA supernatants were able to significantly inhibit zoospore release with IC_50_ values of 2.71% for S38 and 3.45% for Buz14 (non-significant difference). In addition, they did not significantly differ from those obtained against *G. bidwellii* fungal growth.

In addition, LB medium alone was also able to induce a relatively strong inhibition of zoospore release with an IC_50_ of 14.9%, although significantly higher than BCA supernatants. Interestingly, the IC_50_ of LB medium against *P. viticola* is significantly lower than that against the two *G. bidwellii* strains (Table 1).

*In vitro* direct effect against *P. viticola*

The direct effects of treatments were assessed *in vitro* by the determination of their ability to inhibit the release of *P. viticola* zoospores from sporangia and to inhibit the mobility of released zoospores (Figure 7).

As shown in Figure 7A, LB medium, S38 and Buz14 supernatants sharply inhibited the release of zoospores, compared to the water control (more than 90% full sporangia and less than 10%, respectively) (Figure 7A).

Regarding the mobility of zoospores, LB used at 25% did not seem to have any marked direct effect compared to the control (Figure 7B). Conversely, both bacterial supernatants completely inhibited the mobility of zoospores, demonstrating the antibiosis effect of the two bacterial strain supernatants.

### 3.3. Accumulation of Gene Transcripts (STS and ROMT) Contributing to Defence-Related Responses

#### 3.3.1. Gene Expression Analyses by qRT-PCR

To assess the ability of bacterial supernatants to induce defence responses, we focused on stilbene compounds. These compounds are well known to be major compounds in grapevine defence. They act as phytoalexins as they are synthesized *de novo* in response to microbial pathogens and/or abiotic stress, and as they rapidly accumulate at the areas/sites of infection.

First, the accumulation of transcripts of two genes encoding stilbene synthase (STS) and *trans*-Resveratrol di-*O*-MethylTransferase (ROMT) was analysed. As shown in Figure 8, the transcripts of the two defence genes were accumulated in response to LB medium, S38 and Buz14 treatments, compared to water control. LB medium seemed to induce the highest fold change (120.5 and 73.9 for *STS* and *ROMT*, respectively) and Buz14 seemed to be more active than S38. However, no statistical difference was found between the three treatments.

#### 3.3.2. Stilbene Analyses

The capacity of Buz14 and S38 supernatants (25%) to induce stilbenes was analysed by UHPLC-QqQ in leaves of Marselan plants and compared to the controls (water as negative control and LB medium). The amount of 15 stilbenes was measured, including *trans*-resveratrol, *cis*-piceid, *trans*-piceid, piceid isomer, astringin isomer 1, astringin isomer 2, isorhapontin, pterostilbene, *trans*-δ-viniferin, *trans*-piceatannol, piceatannol isomer, *trans*-ɛ-viniferin, viniferin isomer 1, viniferin isomer 2 and *trans*-ω-viniferin. The changes in the leaf stilbene content, determined at 24 h post-treatment, are shown using a principal component analysis (PCA—Figure 9).

The first two main axes accounted together for 70.2% of the total variance, i.e., 56.2% and 14% for PC1 and PC2, respectively (Figure 9A). Water-control samples were clearly separated from the other three treatments, namely LB medium, S38 and Buz14. In addition, both supernatant samples—which cannot be distinguished from each other—were clearly different from the LB medium alone. The PCA (Figure 9B) also allowed us to distinguish the control (water) from LB medium, S38 and Buz14, with stilbenes representative of the treated leaf samples. Nevertheless, three stilbenes did not take part in this separation: *trans*-piceatannol, one of its isomers and an isomer of piceid.

The content of each stilbene studied is shown in Figure 10, except for *trans*-piceatannol, one of its isomers, and an isomer of piceid, since their amount was similar in the different conditions. The LB medium alone significantly increased the content of the 12 stilbenes when compared to the water-control (except for *trans*-piceid). S38 induced a significant increase in the content of eight stilbenes (*trans*-resveratrol, *cis*-piceid, astringin isomer 2, isorhapontin, pterostilbene, *trans*-ɛ-viniferin, viniferin isomer 1 and *trans*-ω-viniferin), and Buz14 in the content of six stilbenes (*trans*-resveratrol, astringin isomer 2, isorhapontin, pterostilbene, *trans*-ɛ-viniferin and *trans*-ω-viniferin).

The LB medium was the treatment leading to the highest accumulation of most stilbenes. More precisely, the content in *trans*-resveratrol, astringin isomer 1, astringin isomer 2, pterostilbene, *trans*-δ-viniferin, viniferin isomer 1 and viniferin isomer 2 were 10.1, 4, 2.4, 2.7, 13.1, 5.2 and 6-fold higher than in the water control condition, respectively. Buz14 was the treatment triggering the greatest accumulation of *trans*-ɛ-viniferin and *trans*-ω-viniferin, 13.6 and 11.2 times more than in the control, respectively. Additionally, S38 resulted in a higher content of isorhapontin compared to LB medium and Buz14, which was 13.3 times more than the control (water). Finally, the amount of *cis*-piceid was equally higher for both LB medium and S38 treatments (and very close to the one of Buz14) than in the control, with a 1.5-fold higher content.

## 4. Discussion

### 4.1. In Planta Multi-Pathogen Preventive Protection

As for the overall protection efficacy, the original results in this study substantiate the broad-spectrum biocontrol ability of the two *Bacillus* strains tested, *B. velezensis* Buz14 and *B. ginsengihumi* S38. Most published papers in the biocontrol field using *Bacillus* strains as BCA, particularly addressing post-harvest pathosystems, focused on very few pathogens, mostly just one [63]. This may have resulted from both (i) the important biocontrol stepwise screening strategy theorized by [64] (widely accepted paper with 170 citations), and (ii) the time-consuming *in vivo* screening biotests. In this study, for the first time against two key grapevine diseases, namely DM and BR, *in planta* biotests showed a high and significant biocontrol efficacy of both *Bacillus* supernatants tested. In terms of broad-spectrum disease control, the Buz14 strain was then further confirmed as showing a very high multi-pathogen antifungal capacity, including now in its BCA spectrum a biotrophic (*P. viticola*) and an hemibiotrophic pathogen (*G. bidwellii*). This is in addition to the postharvest ones, mostly necrotrophs, screened initially and including *Penicillium digitatum*, *P. expansum*, *P. italicum*, *Monilinia fructicola*, *M. laxa* and *B. cinerea* [49]. Comparatively, the S38 strain had not been shown with such a broad multi-pathogen antifungal capacity, exhibiting an efficient antagonistic effect against Botrytis bunch rot [41,48] but a limited efficacy against two necrotrophic pathogens, *N. parvum* and *P. chlamydospora* [37,50].

Concerning *G. bidwellii*, very importantly and interestingly, the biocontrol antifungal efficacy has been demonstrated in this paper against the two major types of BR infections, primary and secondary ones caused by sexual (ascospores) and asexual (pycniospores) inocula, respectively. In 24 h preventive treatments before inoculations by either pycniospores or ascospores, both BCA supernatants exhibited high efficacy as biocontrol active substances, at the concentration of 50%. Likewise, but at a lower concentration (25%), this was clear for pycniospore infections (Figure 1). Furthermore, both supernatants were also highly effective against *P. viticola*. This is the first efficacy demonstration by using a strain of the *Bacillus* species *ginsengihumi* (S38). As for fungal BCA, [65] reported the efficacy of *Trichoderma harzanium*, but only a few BCA effective against DM have been described in the literature. Regarding bacterial BCA, it is noticeable that, for example, only one is registered in France, based on *B. amyloliquefaciens* (strain FZB24) [66]. In the literature, three endophytic strains were isolated from grapevines: two from leaves, i.e., the *B. subtilis* strain GLB191 [67] and the *B. pumilus* strain GLB197 [67,68], and the *B. velezensis* strain KOF112 from shoot xylem [69]. *B. ginsengihumi* S38 is also an endophytic bacterium isolated within the grapevine wood tissue [37]. This suggests that endophyte bacteria might have a higher potential to control *P. viticola* than epiphytic ones. Accordingly, ref. [70] observed that inhibition of *Phytophthora infestans* was achieved by strains of several genera, more frequently among endophytes than epiphytes. This was based on 78 bacteria isolated from foliar endophytic and epiphytic communities, in three grapevine cultivars, and tested against the oomycete *P. infestans* (as a surrogate for the obligate parasite oomycete *P. viticola*). Thus, endophytic bacteria may be a more promising resource than epiphytic bacteria for identifying BCA active against DM. In addition, the fact that culture filtrates or supernatants are also active (results obtained in this study and in [33]) facilitates their use in the vineyard, as the efficacy does not depend on the bacterial colonization and/or survival capacity and response of the microorganism to the environment, but only on the active substances they produce.

### 4.2. Characterisation of the Antibiosis Mode of Action

Against *G. bidwellii*, (Figure 4), the antibiosis mode of action of *B. velezensis* Buz14 was clearly demonstrated *in planta*, at the supernatant concentration of 25%, by the 2 h preventive treatments that were efficient significantly against ascospore infections (whatever the cultivar, Marselan or Artaban). Thus, direct antibiosis and toxic effects of the bacterial Buz14 supernatant were shown on the ascospore inoculum of BR. Because of the very short 2 h preventive duration, it is indeed not hypothesized that the supernatant may have elicited the grapevine defences, hence excluding this mechanism for contributing significantly to the observed biocontrol efficacy. However, we may put forward two potential direct effects, possibly acting together: (i) a surface physical effect, i.e., interfering with and impeding the early step of spore attachment at the leaf surface, and (ii) the antibiosis effect affecting spore germination, mycelial growth and/or appressorium formation. Since the anti-germinative effect was not demonstrated on pycniospores from the “GF2” BR strain, the latter assumption for the germination should be further investigated, in the near future, by testing *in vitro* ascospore germination and associated appressorium formation. Nonetheless, it was previously demonstrated that both *G. bidwellii* spore types displayed similar germination and infection patterns [12]. Thus, the pattern observed for pycniospores should be similar regarding ascospores. In addition, it is interesting to note that in the presence of the supernatants, pycniospores appeared misshapen. Hence, even though the germination rates were not affected by the supernatants, spore viability might have been affected, which should be further investigated. Similarly, the first hypothesis of leaf surface interference also needs to be further studied because such a mode of action has already been demonstrated as quite effective against the BR pathogen [13]. By studying different saponin-containing extracts, with an efficacy of more than 90%, such biocontrol molecules affecting key leaf surface properties (notably hydrophobicity) were indeed shown as very efficient against the BR pathogen [13]. These molecules, known as powerful surfactant compounds, were issued from *Sapindus mukorossi*, *Chenopodium quinoa* and *Quillaja* sp. As for *G. bidwellii*, the antibiosis was also shown *in vitro*, conclusively, regarding both BCA strains via the mycelial growth tests, resulting in low IC_50_ values that were always lower than 5.7% (Table 1). In addition, very similar results were obtained for the two supernatants in comparison with Sonata^®^, a marketed biocontrol product in France based on *B. pumilus* (QST 2808) with an IC_50_ of about 1.6%. Among the very few studies showing the potential of *Bacillus* BCA against *G. bidwellii*, an organically registered biofungicide, based on *B. subtilis* (Serenade^®^), has been reported recently in Virginia [46]. However, with the same biocontrol product, a previous study showed the crucial role of the formulation in the efficacy of the registered product [45].

This antibiosis mode of action of both Buz14 and S38 supernatants was also clearly demonstrated for *P. viticola*. *In vitro*, both supernatants inhibited the release of zoospores from sporangia and the zoospore mobility. They displayed similar activities, with IC_50_ values of 2.7 and 3.4% for S38 and Buz14 strains, respectively. Accordingly, the application of the supernatants on leaves, two hours before inoculation (time too short for eliciting significant defence responses), led to a strong reduction in the number of infection sites, i.e., encysted zoospores. As mentioned above, it could be due to surface physical effects impairing zoospores to reach stomata and/or to antibiosis affecting zoospore mobility and viability as observed *in vitro*. This needs to be further investigated to check if both effects may be involved. *In planta*, the antibiosis effect of culture filtrates of *B. subtilis* strain GLB191 were also shown against *P. viticola* [33]. Another published article also demonstrated the antibiosis effects by the *B. velezensis* strain KOF112 against the oomycete *P. infestans* [69].

Very interestingly in the broad-spectrum BCA prospect, we found that both BCA supernatants displayed a similar efficacy in terms of IC_50_ regarding the fungal growth inhibition of *G. bidwellii* and the zoospore-release inhibition of *P. viticola*. Thus, this substantiates the broad-spectrum biocontrol ability of both BCA supernatants and may be indicative of the potential use of such active substances, with a similar dosage, against the two key grapevine pathogens in further studies. Such a great antibiosis capacity of the two BCA model strains was also reported in several other studies dealing with various strains within the *Bacillus* genus that could be used as powerful biocontrol products based on viable BCA and/or their active substances. Acting via various and complementary modes of action, such as antagonism and/or plant defence induction following the production of key metabolites, and/or nutrient and space competition, they indeed represent a promising tool for crop protection [33,71]. More specifically, they are known to produce different cyclic lipopeptides belonging to the following families: surfactins, iturins, fengycins, kurstakins and locillomycins [71,72]. Different *B. velezensis* isolates were previously reported as valuable BCA owing to the nature of the cyclic lipopeptides they secrete, mostly identified as belonging to the surfactins, iturins and fengycin families [48,71,73]. In that regard, the *B. velezensis* Buz14 strain antifungal activity was demonstrated against several postharvest moulds, including *B. cinerea*, and mostly attributed to hydrosoluble metabolites [73,74]. Fengycin corresponded to the most abundant compound (853 µg.mL^−1^) in the lipopeptide fraction, while Iturin A, despite a lower concentration (407 µg.mL^−1^), presented a stronger antifungal activity [73]. Interestingly, *B. velezensis* strains could also act through the synthesis of polyketides [48,72,75] as well as the release of volatile compounds displaying antifungal properties [72,75]. This is also documented in the *B. velezensis* Buz14 strain, since up to 15 volatile metabolites were identified, including benzaldehyde and diacetyl as the most effective [75]. Furthermore, it was concluded that the living cells of these *Bacillus* strains did not account for the key and main part of the biological activity, compared with the bacterial culture supernatants [48].

By contrast, very few articles deal with *B. ginsengihumi* strains. As other *Bacillus* spp., they are capable of producing cyclic lipopeptides, but their nature and concentration are to date unknown. They could also possibly act through the synthesis of siderophores, known to interfere with the development of various microorganisms [72,76,77]. Some strains are also described for their ability to produce a phytohormone, indolacetic acid, so as to promote plant growth, alongside fungicidal metabolites [76].

Therefore, based on such a well-known synthesis of various biologically active compounds [37,47,78], our results further substantiate the fundamental role of the *Bacillus* genus in sourcing for new BCA strains inhibiting various species of plant pathogens. This seems to us very important to underline such a sourcing for an actual BCA broad-spectrum prospect. Further studies are underway to better elucidate the active molecules that are present in the supernatants produced by these two model BCA strains. Genome sequencing approaches have also been initiated to forecast their potential production, notably of lipopeptide and/or polyketide antifungal compounds.

### 4.3. Plant Defence Elicitation

Buz14 and S38 supernatants (i.e., LB medium containing active metabolites released by bacteria) induced the expression of two defence genes, *STS* and *ROMT*, concomitantly with the synthesis of several stilbenes (Figure 8 and Figure 10). Both S38 and Buz14 increased significantly the content of six stilbenes: *trans*-resveratrol, astringin isomer 2, isorhapontin, pterostilbene, *trans*-ɛ-viniferin and *trans*-ω-viniferin. The targeted genes *STS* (stilbene synthase) and *ROMT* (resveratrol *O*-methyl transferase) are involved in the synthesis of the stilbene phytoalexins, resveratrol—the monomer unit of all stilbenes—and pterostilbene, respectively. In response to disease or injury stresses, it is well recognized that an active resistance mechanism is set up involving particularly the up-regulation of the enzyme STS. The consecutive formed compounds, resveratrol and its derivatives, including viniferins (resveratrol dimers or tetramers) and pterostilbene (dimethylated form of resveratrol), are known to have high antimicrobial activity against grapevine pathogens, including *P. viticola* [79]. Resveratrol inhibits the germination of sporangia of *P. viticola*, and a mixture of stilbenes has also been shown to affect zoospore mobility [80,81]. Furthermore, pterostilbene displays a higher fungal inhibitory activity compared to resveratrol, and the methylation of hydroxyphenyl groups would be responsible for this better biocidal activity [80]. In addition to pterostilbene, viniferins are also considered as highly effective compounds against DM [80,82]. Thus, such significant accumulations of the most toxic forms of stilbenes in grapevine leaves treated by the bacterial supernatants can explain, to a certain extent, their enhanced resistance to *P. viticola* and *G. bidwellii.* Very interestingly and for the first time, we thus demonstrated the ability of the supernatants of *B. velezensis* Buz14 and *B. ginsengihumi* S38 to stimulate stilbene accumulation both at the gene expression level and in terms of metabolite contents in grapevines. Although this important biocontrol mode of action had not been investigated in these two strains, other *Bacillus* BCA strains such as *B. velezensis* and *B. amyloliquefaciens* were known for activating key defence-related enzymes in different plant host species, including grape, tomato and loquat [69,83,84]. The activation of grapevine defences by the culture filtrate of *B. subtilis* strain GLB191 were also demonstrated, together with an antibiosis effect [33]. The generation of mutants, affected in the production of fengycin and/or surfactin, allowed them to conclude that activity was partly due to these lipopeptides. Similarly, our results suggest that compounds released by bacteria in their culture medium show both antibiosis and defence eliciting marked anti-pathogen effects.

### 4.4. Antifungal Activity of the LB Culture Medium

In this study, by itself, the LB medium also showed partial effects against both pathogens and induced plant defences. Similar effects were observed with the PDB (Potato Dextrose Broth) bacterial culture medium [33]. So, for the bacterial supernatants we have assessed, the question arises of the origin of the observed activity and the relative complementary parts played by the initial components of the LB medium and by the bacteria-secreted metabolites (as discussed above). The LB medium showed an antimicrobial effect against both *P. viticola* and *G. bidwellii*; the effectiveness was much lower than that of the supernatants. First, our results clearly showed a direct antibiosis effect of the LB medium alone, on both pathogens, with IC_50_ in the range of approx. 15–40%, for the two pathogenic strains infecting *V. vinifera* (Table 1). As for *G. bidwellii*, such results of a direct effect on the mycelial growth have not been published previously, and further studies are then needed to investigate some of the possible reasons. Accordingly, in the *in planta* bio-tests at the 25% concentration (Figure 4), the LB medium by itself, applied two hours before inoculation by ascospores, also evidenced an unexpected, yet significant antifungal effect. At this 25% concentration, it also significantly reduced both the release of *P. viticola* zoospores and the number of encysted zoospores, although it did not affect their mobility as did supernatants. As previously reported for PDB against *P. viticola* [33], both effects were noted by reducing the number of infected sites (compared to the water control) and by activating plant defences. Second, LB medium at 25% significantly induced the expression of defence genes, compared to the water control, at a quite similar level as bacterial supernatants. LB medium contains yeast extracts (LB product sheet), which have been reported as plant defence inducers. It was, for instance, used to induce the accumulation of phenolic compounds in flowering flax (*Linum grandiflorum* Desf.) cells cultured *in vitro* [85]. Similarly, application of a yeast extract to *Zataria multiflora* cell suspensions induced the production of H_2_O_2_, NO, malondialdehyde, phenylpropanoids and terpenoids [86]. Yeast extracts also contain vitamins, among which are B vitamins (product data sheet). It was shown that riboflavin (vitamin B2) and thiamine (vitamin B) induced grapevine defence and resistance to DM [87,88]. Due to this activity of the LB medium by itself, this control shows also its own limitations. It is not a perfect experimental control since, during bacterial cultivation, it is indeed depleted in constituents used by bacteria for their growth, multiplication and metabolism, conversely to the non-inoculated one.

Overall, we can conclude that the supernatants of *B. velezensis* Buz14 and *B. ginsengihumi* S38 (LB medium plus metabolites secreted by bacteria) have a dual mode of action associating both antibiosis and elicitation of the plant defences.

## 5. Conclusions

This study highlights the effectiveness of two bacterial culture supernatants, from *B. velezensis* Buz14 and *B. ginsengihumi* S38 strains, for bioprotection against the grapevine key diseases, downy mildew and black rot. Previous work has also shown them to be effective against grey mould and various other necrotrophic pathogens. These supernatants therefore have a greatly interesting broad-spectrum antagonistic action for grapevine protection against microorganisms of very different lifestyles, i.e., biotrophic, hemibiotrophic and necrotrophic. Furthermore, their mode of action is demonstrated as clearly dual, combining both antibiosis and stimulation of plant defences. Further investigation is needed to assess their efficiency in the vineyard and to identify the active bacterial compounds. It would also be interesting to further investigate their effectiveness against other grapevine pathogens and key diseases, such as powdery mildew, as well as those in other major crops.

## Figures and Tables

**Figure 1 jof-10-00471-f001:**
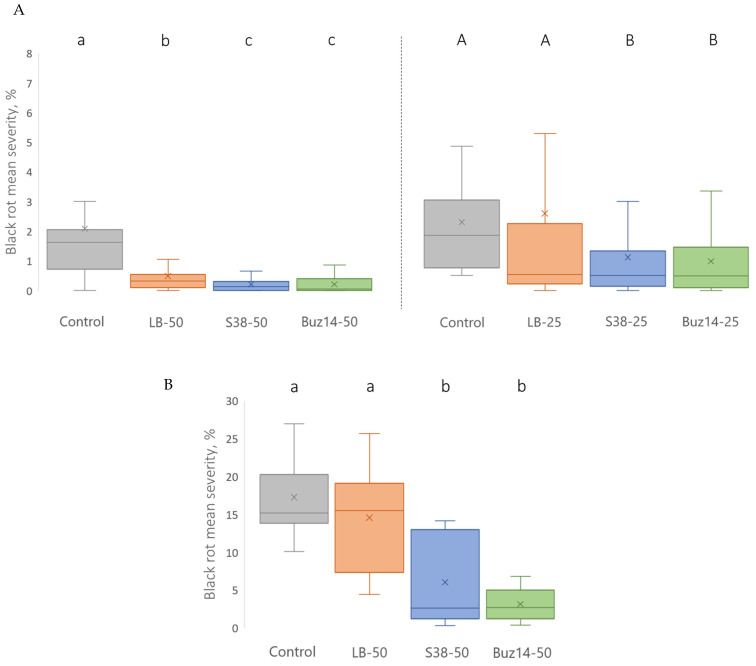
BCA supernatant preventive efficacy against *G. bidwellii* foliar symptoms originating from pycniospore-based (**A**) and ascospore-based (**B**) inocula, when applied 24 h pre-inoculation. Results are means ± SD (n = 27 and n = 8, resp.). Means followed by the same either lowercase or capital letter are not significantly different, by two-way and one-way ANOVA (α = 0.05) tests, respectively.

**Figure 2 jof-10-00471-f002:**
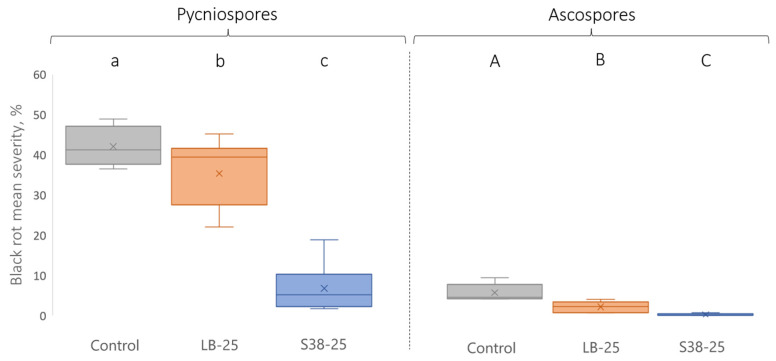
BCA supernatant preventive efficacy against *G. bidwellii* foliar symptoms originating from both pycniospore- and ascospore-based inocula, when applied 48 h pre-inoculation. Results are means ± SD (n = 6). Means followed by the same lowercase or capital letter are not significantly different, by two-way ANOVA (α = 0.05).

**Figure 3 jof-10-00471-f003:**
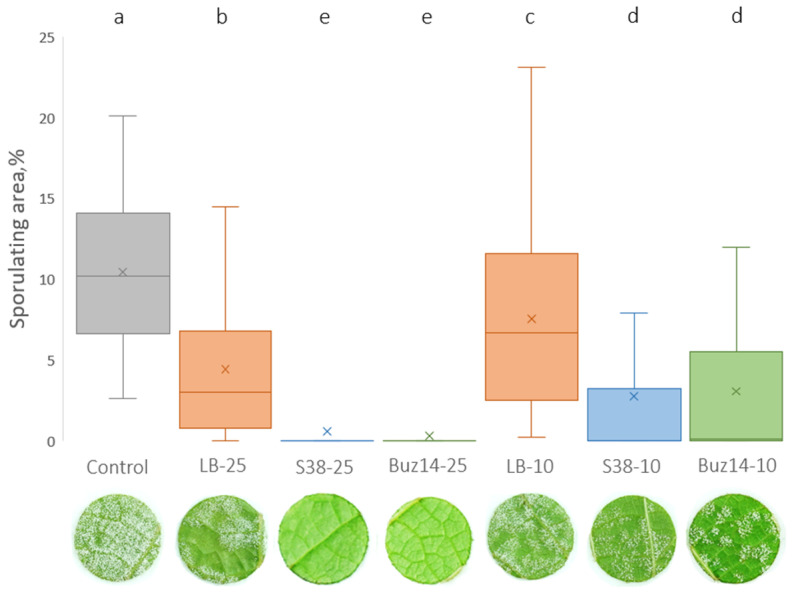
Effect of BCA supernatants on *P. viticola* sporulation. Leaves of cv. Marselan were sprayed with water (control) or LB medium, Buz14, and S38 supernatants at 10% and 25%. *P. viticola* was inoculated 48 h post-treatment, and sporulation was evaluated at 7 days post-inoculation. Results are means ± SD (n = 3). Means followed by the same letter are not significantly different, by the non-parametric Kruskal–Wallis test and pairwise Wilcoxon post-hoc test (α = 0.05). Photographs correspond to representative leaf discs.

**Figure 4 jof-10-00471-f004:**
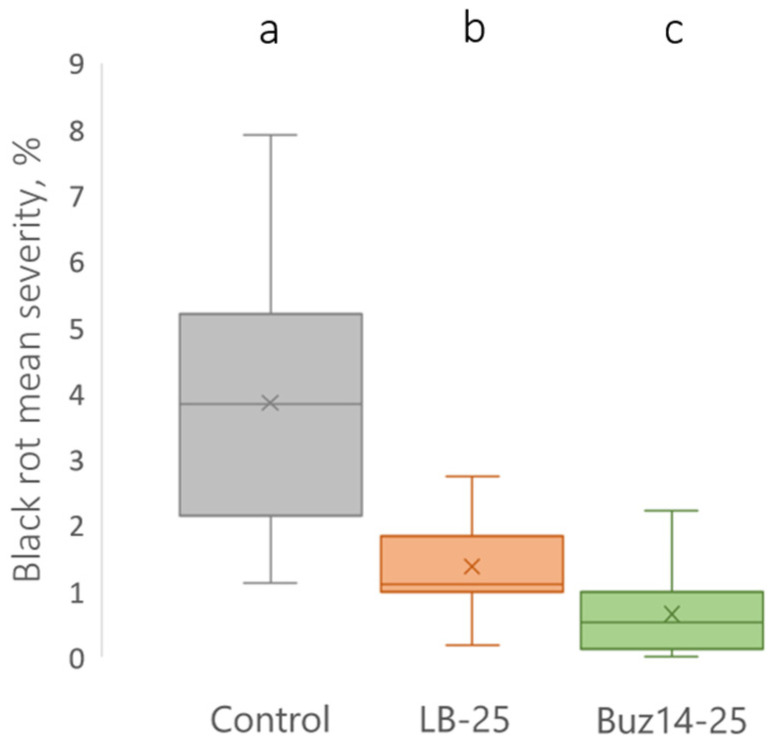
BCA supernatant *in planta* direct effect against *G. bidwellii* foliar symptoms originating from an ascospore-based inoculum, when applied 2 h pre-inoculation. Results are means ± SD (n = 14). Means followed by the same letter are not significantly different, by two-way ANOVA (α = 0.05).

**Figure 5 jof-10-00471-f005:**
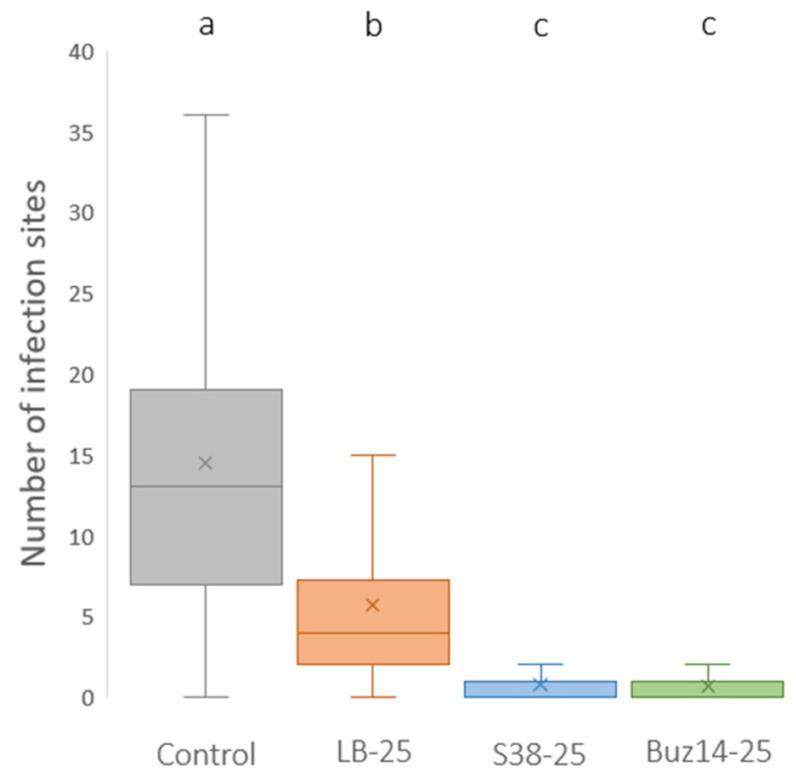
*In planta* direct effect of BCA supernatants against *P. viticola*. Leaves (cv. Marselan) were sprayed with water (control) or LB medium, S38, and Buz14 supernatants at 25%. Inoculation was performed 2 h post-treatment. Observation by epifluorescence microscopy was performed 24 h post-inoculation, and infection sites (i.e., encysted zoospores) were counted. Results are means ± SD (n = 3). Means followed by the same letter are not significantly different, by the non-parametric Kruskal–Wallis test and pairwise Wilcoxon post-hoc test (α = 0.05).

**Figure 6 jof-10-00471-f006:**
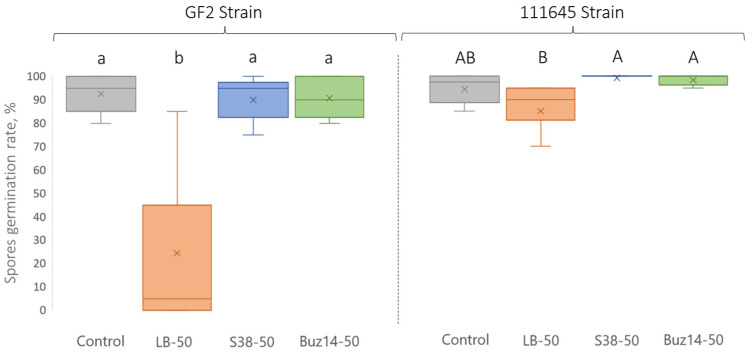
*G. bidwellii* pycniospore germination rates determined 16 h after inoculation. Values are means ± SD (n = 8). Means followed by the same lowercase or capital letter are not significantly different, by one-way ANOVA comparison (α = 0.05).

**Figure 7 jof-10-00471-f007:**
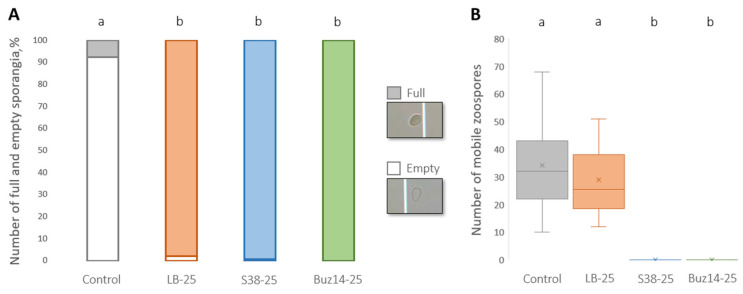
*In vitro* direct effect against *P. viticola.* (**A**) Percentage of empty and full *P. viticola* sporangia observed 2 h after treatment. (**B**) Number of mobile *P. viticola* zoospores observed 2 min post-treatment. Treatments: water (control), LB medium, S38 and Buz14 supernatants at 25%. Values are means ± SD (n = 9). Means followed by the same letter are not significantly different, by the non-parametric Kruskal–Wallis test and pairwise Wilcoxon post-hoc test (α = 0.05).

**Figure 8 jof-10-00471-f008:**
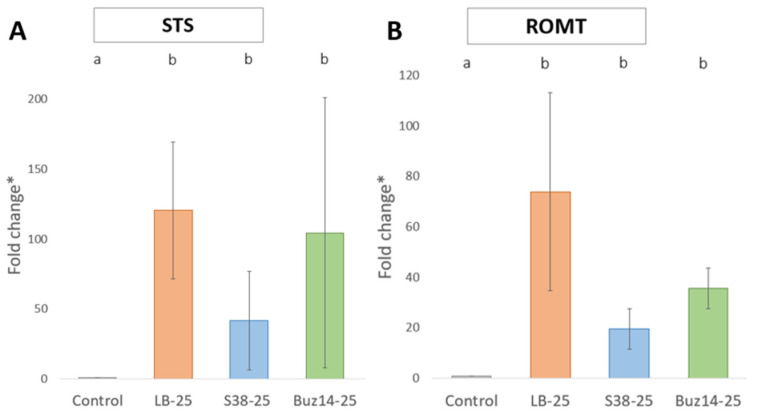
Relative expression of defence-related genes in grapevine leaves. Leaves of cv. Marselan were sprayed with water (as control), LB medium, S38 or Buz14 supernatants at 25% and collected at 10 hpt. Means (n = 9) followed by the same letter are not significantly different by parametric ANOVA test and Tukey post-hoc test (α = 0.05). * Fold change in gene expression was calculated with the Common Base Method (fold change = 10 −ΔΔC(w)q). ROMT: *trans*-resveratrol di-*O*-methyltransferase; STS: stilbene synthase.

**Figure 9 jof-10-00471-f009:**
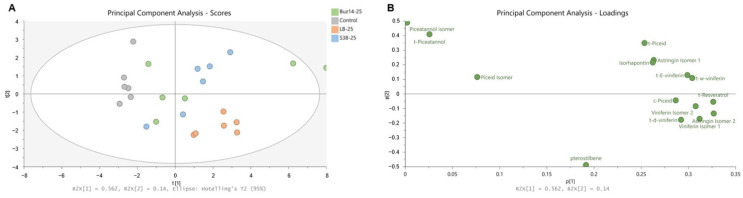
Principal component analysis (PCA) on all stilbenes quantified (15 in total) in treated grapevine leaves. (**A**) PCA score plot; (**B**) PCA loading plot.

**Figure 10 jof-10-00471-f010:**
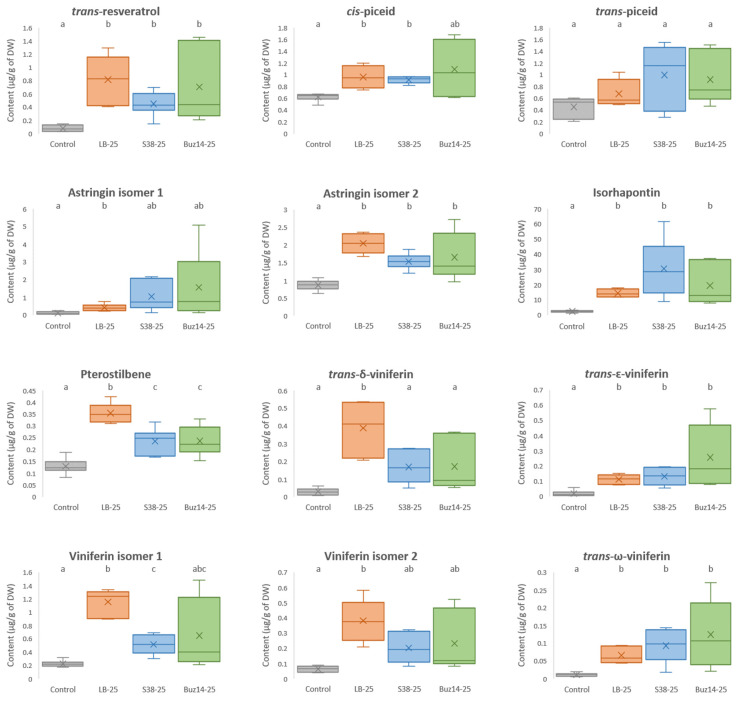
Stilbene content of the 12 stilbenes studied in grapevine leaves. Leaves of cv. Marselan were sprayed with water (as control), LB medium, S38 or Buz14 supernatants at 25% and collected at 24 hpt. Means followed by the same letter are not significantly different by parametric ANOVA test or non-parametric Kruskal–Wallis and Tukey post-hoc test or pairwise Wilcoxon post-hoc test (α = 0.05).

**Table 1 jof-10-00471-t001:** Supernatants and LB medium IC_50_ values (%) arising from the antifungal *in vitro* direct contact bioassay against *G. bidwellii* and *P. viticola*.

Pathogen	Product IC_50_ Value (%)		
	LB Medium	S38	Buz14
*G. bidwellii* GF2	39.8 ± 11.9 a	3.6 ± 3.7 b	5.7 ± 3.2 b
*G. bidwellii* 111645	64.1 ± 14.6 c	1.3 ± 0.3 b	1.8 ± 0.8 b
*P. viticola*	14.9 ± 0.7 d	2.71 ± 0.5 b	3.45 ± 0.2 b

Values are means ± SD (*G. bidwellii* n = 5 and *P. viticola* n = 4). Means followed by the same lowercase letter are not significantly different, by two-way ANOVA comparison (α = 0.05).

## Data Availability

The raw data supporting the conclusions of this article will be made available on request from the corresponding author.

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
