# Peer review of "Broad-Spectrum Efficacy and Modes of Action of Two *Bacillus* Strains against Grapevine Black Rot and Downy Mildew"

_jof, 2024, doi:10.3390/jof10070471_

Round 1

Reviewer 1 Report

Comments

In abstract and introduction

First: Strongly indicate why to consider the two Bacillus strains, 115 Buz14 and S38 in the study, make a difference, other way it seems to be a “me too” study.

Second. Indicate clearly why to include LB as a treatment. No reason is given, and then it appears in methodology and results as a prominent component of the study.

Methodology

In section 2.2.1. Black rot experimental design and disease evaluation

Two concentrations are used, 25 and 50%, however, it is not clearly indicated when one or other was used. Indicate the criteria.

Results

Regarding the results for the point 2.2.1 in methodology. It seems that the two concentrations were used, but in methodology it reads “25 or 50”. It should read “25 and 50”.

Discussion

Subtitles may help to follow the discussion.

General comment

The paper is well written, only minor corrections are needed.

See comments in MS.

See comments in MS

Author Response

Dear reviewer, thank you for your evaluation of our manuscript and your relevant comments. Please see the attachment for a detailed response to all your comments.

Reviewer 2 Report

Control effects of two bacterial BCAs are shown towards vine diseases caused by a fungus and by an oomycete. The graphs are informative but photos would help reader understand the results better. More details are needed for the methods especially whether the supernatant can function as an inoculum when applied to plants, and the composition of LB and preparation in autoclave etc.

The titles of results sub-sections should describe the main finding of that subsection and not just repeat the method used.

For figure legends, write how many times the experiment was repeated and whether the trends were the same in repeat experiments.

It is difficult to make inferences about the LB medium because the composition of the non-inoculated medium will be different from that after the bacteria have metabolized its components.

Curative applications of BCAs? Was this attempted?

Do the authors have any patents for their BCAs?

What other studies have shown similar effects of LB medium? The authors cite a study about PDA in the discussion.

Ln 133: expanded?

Ln 168: humidity chamber?

Ln 176: what are the ODs of the cultures at the time of harvesting?

Ln 177: Write the recipe for LB. There is more than one way to make LB medium. Did you try to but from another company?

Ln 178: Was a filter membrane used to ensure the supernatant was cell free? Centrifugation is not enough for this cell-free (sterile) claim. Are the bacteria growing again when applied to the plants or is it only due to the effects of compounds produced in the previous culture.

Ln 407: the protection rate of 97% with ascospores is a bit misleading because the disease severity is quite low (there is not much disease to protect against).

Ln 544: Why focus on stilbenes?

Ln 738: “Buz14 and S38 supernatants induced the expression of two defense genes,” these type of claims are difficult to make if the LB medium also induced the genes.

Can representative pictures of plants be shown for Fig. 1 and Fig. 2?

Fig. 8: Can error bars be added to graphs?

Author Response

(The authors gave the same response as above.)

Reviewer 3 Report

The presented article by Robin Raveau et al., entitled “Broad-spectrum efficacy and modes of action of two Bacillus strains against grapevine black-rot and downy mildew” is relevant and aimed to solve very important problem: Grapevine protection from the most harmful diseases using eco-friendly approach (i.e. microbial antagonists Bacillus spp.). The study provides valuable information about the biocontrol potential and mode of action of bacterial culture supernatants from B. velezensis Buz14 and B. ginsengihumi S38 against black rot (Guignardia bidwellii) and downy mildew (Plasmopara viticola) via antibiosis and induction of plant defense mechanisms (particularly by regulating stilbenes production). In addition, the authors demonstrated the partial effects of LB medium, used for the bacterial cultures, against both pathogens and plant defense induction. In general, the obtained results are well presented and discussed. The results are valuable in both fundamental and practical aspects and will be of interest to a broad range of readers.

However, I recommend to include Conclusion part as a separate section with a brief and quite detailed description of the obtained results highlighting the novelty. It will help readers to perceive the information easier.

Best wishes

Some minor text editing (formatting) is required.

Author Response

Dear reviewer, thank you for your evaluation of our manuscript and your relevant comments. Please see the attachment for a detailed response to your comments.

Reviewer 4 Report

The manuscript "Broad-spectrum efficacy and modes of action of two Bacillus strains against grapevine black-rot and downy mildew" made a very positive impression from a scientific point of view.

The manuscript presents the results of the development of a method for protection from phytopathogens based on biologically active substances produced by bacteria. The relevance, purpose and tasks are well and thoroughly explained in the introduction.

Research methods and organisms are also described in sufficient detail. They include a comprehensive approach to research. However, this section does not contain enough information on the organisms . The authors should characterize the strains of microorganisms in more detail. Is there a number in the genbank? Are the strains deposited in collections? If not, how was the identification performed?

The results are presented in great detail and visually. The authors need to improve the figures by removing the horizontal grid lines, as well as replace "(%)" with ",%" in the axis caption. Figure 8 does not explain what * is.

The discussion section is very informative. I would like to mention a large list of references, which indicates that the authors possess information about the research problem.

Author Response

(The authors gave the same response as above.)

Round 2

Reviewer 2 Report

The authors made valuable improvements to the manuscript. 

It would be useful for reader if the authors made the reviewer comments and their replies "open" and published alongside the manuscript.

as by the below comment, the pictures of plants should be included as supporting files.

Can representative pictures of plants be shown for Fig. 1 and Fig. 2? RESPONSE: You are indeed right, pictures of plants could be very useful to illustrate the results that we obtained. However, it is very difficult to take whole-plants pictures that are representative, since grapevine leaves affected by black rot are most of the time not visible within a same plan. In addition, we chose not to include pictures in the main text, since the number of figures is already quite high. If you find it particularly relevant, we could nonetheless provide pictures of the leaves affected by black rot in our conditions, as supplementary material. 

see above
